# Synthesis and Properties of Ferrite-Based Nanoparticles

**DOI:** 10.3390/nano9081079

**Published:** 2019-07-27

**Authors:** Kayrat K. Kadyrzhanov, Kamila Egizbek, Artem L. Kozlovskiy, Maxim V. Zdorovets

**Affiliations:** 1Engineering Profile Laboratory, L.N. Gumilyov Eurasian National University, Astana 010008, Kazakhstan; 2Laboratory of Solid State Physics, The Institute of Nuclear Physics, Almaty 050032, Kazakhstan; 3Laboratory of Additive Technologies, Kazakh-Russian International University, Aktobe 030006, Kazakhstan; 4Department of Intelligent Information Technologies, Ural Federal University, Yekaterinburg 620075, Russia

**Keywords:** ferrite nanoparticles, crystal structure, phase transformations, magnetic structures, nanotechnologies

## Abstract

The work is dedicated to the study of the structural and optical characteristics, as well as the phase transformations, of ferrite nanoparticles of CeO_2_-Fe_2_O_3_. To characterize the results obtained, the methods of scanning and transmission microscopy, X-ray diffraction (XRD) spectroscopy, and Mössbauer spectroscopy were applied. It was found that the initial nanoparticles are polycrystalline structures based on cerium oxide with the presence of X-ray amorphous inclusions in the structure, which are characteristic of iron oxide. The study determined the dynamics of phase and structural transformations, as well as the appearance of a magnetic texture depending on the annealing temperature. According to the Mossbauer spectroscopy data, it has been established that a rise in the annealing temperature gives rise to an ordering of the magnetic properties and a decrease in the concentration of cationic and vacancy defects in the structure. During the life test of synthesized nanoparticles as cathode materials for lithium-ion batteries, the dependences of the cathode lifetime on the phase composition of nanoparticles were established. It is established that the appearance of a magnetic component in the structure result in a growth in the resource lifetime and the number of operating cycles. The results show the prospects of using these nanoparticles as the basis for lithium-ion batteries, and the simplicity of synthesis and the ability to control phase transformations opens up the possibility of scalable production of these nanoparticles for cathode materials.

## 1. Introduction

Today, there is huge interest in the synthesis of new materials used as the basis for the creation of photocatalysts, anode materials for lithium-ion batteries, microelectronic devices, etc. [1,2]. Among the variety of methods for obtaining new materials, of particular interest are methods that allow obtaining nanostructured materials, the interest in which is due to their structural properties, as well as a large area of active surface [3,4]. However, despite the enormous number of research works on methods of obtaining nanostructures, interest in studying their properties, the impact of various factors on the alteration in their structural and dimensional characteristics, as well as the effect of structure and phase composition on the practical application of these nanomaterials has not yet dried up, but only increases every day. The impetus for this is the lack of full information about the behavior of materials and the change in phase composition on a nanoscale [5,6,7,8]. This fact pushes all new groups of researchers to search for new materials, as well as the study of their structural features and the search for practical applications [9,10,11,12,13].

One of the promising candidates in modern alternative energy as the basis of solid fuel elements are perovskite-like systems based on RFeO_3_ ferrite (R = Ce, Y, Ba, Sr) [14,15,16,17,18]. The increased interest in this class of structures is due to their distorted crystal structure with the possibility of filling vacancies with equivalent iron ions [19], which has an essential impact on the structural, optical, magnetic, and dielectric properties [20,21,22,23]. However, the presence of magnetic domains in the structure of ferrite systems allows using them as magneto optical sensors, catalysts, and devices for long-term data storage [24,25]. As is known, nanoscale structures, in contrast to macrostructures, have a large active surface area, higher chemical and corrosion resistance, and unique magnetic and electrical properties, which led to their active use in modern energy and microelectronics [26,27]. Among the wide variety of oxide perovskites and perovskite-like structures, special attention is paid to ferrites, in which cerium is used as a rare-earth element [28,29,30], due to its crystalline structure and conductive characteristics. As a rule, such structures are obtained by solid-phase synthesis [31,32], hydrolysis [33,34], mechanochemical treatment, followed by heat treatment [35,36], the results of which are micro or macro particles with non-uniform composition. Cerium ferrite is characterized by high corrosion and structural stability, which makes it environmentally safe and suitable for use in almost all branches of solid fuel energy.

Earlier, we obtained systems based on perovskite-like structures of the type A(FeM)O_4−x_ (A = Ce; M = Ti) using the method of mechanical chemical synthesis and successive thermal annealing [37]. Moreover, according to the data obtained, the addition of the second component M = Ti result in the formation of structures with predominant phases containing oxide forms of titanium and its compounds with iron. During the study, it was found that the use of the method of mechanical chemical synthesis and subsequent thermal annealing allows to obtain a multiphase system containing a large number of oxide phases or complex non-stoichiometric compounds, which leads to disordering of the crystalline and magnetic texture. Despite the simplicity and efficiency of the application of the mechanical chemical method of synthesis to obtain perovskite-like structures, this method does not allow to obtain particles of nanoscale diameter with clearly controlled phase and stoichiometric compositions. One way to obtain such particles is to combine methods of chemical synthesis with subsequent thermal annealing of the nanoparticles obtained [38,39].

This article represents the outcome of synthesis and studies of the structural and optical properties of nanoparticles CeO_2_-Fe_2_O_3_, obtained by chemical synthesis and further thermal annealing in an oxygen-containing medium. The distinction of this work from previous studies is the combination of methods of chemical synthesis allowing us to obtain oxide forms of nanoparticles with heat treatment. That enables, changing the annealing temperature, to control phase transformations and crystalline and magnetic properties [38,39,40].

## 2. Materials and Methods

Ce (NO_3_)_3_·6H_2_O, FeCl_3_·6H_2_O, HCl and NaOH (Sigma-Aldrich, St. Louis, MO, USA) were used as the source components for the chemical synthesis of ferrite nanoparticles based on iron and cerium oxides. Ce (NO_3_)_3_·6H_2_O (17.36 g) was dissolved in a solution of 400 mL of dioinized water + 1 mL of HCl, then a mixture of FeCl_3_·6H_2_O (10.8 g) + 400 mL of dioinized water + 1 mL of HCl was added, the solution was aged for 48 h in argon medium, after which the precipitate was separated by centrifuging, and washed and dried in argon medium. The pH control was carried out using a pH meter and brought to a level of 9.3 by adding drops of an alkaline solution of NaOH. At this pH level, the formation of hydroxide inclusions is excluded, which was also confirmed using X-ray phase analysis, according to which hydroxide structures are not observed in the structure of the initial samples. Annealing of synthesized nanoparticles was conducted in an oxygen-containing medium in a muffle furnace at a temperature of 200, 400, 600 and 800 °C for 5 h.

Analysis of changes in structural parameters as a result of synthesis and subsequent annealing was carried out using scanning electron microscopy (SEM, Hitachi TM3030 (Hitachi Ltd., Chiyoda, Tokyo, Japan)), and X-ray diffraction (XRD, D8 ADVANCE ECO Bruker, Karlsruhe, Germany). The study of optical characteristics before and after annealing was carried out using the method of ultraviolet (UV) spectroscopy (Analytic Jena Specord-250 BU (Analytik Jena, Jena, Germany). Diffuse reflectance ultraviolet-visible (UV-Vis) spectra were recorded using an Analytic Jena Specord-250 BU spectrophotometer(Analytik Jena, Jena, Germany) equipped with integrating sphere. BaSO_4_ was used as a standard. The resolution was chosen to be 1 nm and the scan speed was 20 nm/s. The spectral range was from 190 nm to 1100 nm.

The analysis of magnetic characteristics and the dynamics of their changes as a result of annealing was carried out using the Mössbauer spectroscopy method (MS1104Em spectrometer, Chernogolovka, Moscow Oblast, Russia). The Mossbauer spectrometer was calibrated at room temperature using a standard α-Fe absorber. For processing and analyzing the Mossbauer spectra, methods were used to restore the distributions of the hyperfine parameters of the Mossbauer spectrum, taking into account a priori information about the object of study, implemented in the SpectrRelax program [41,42].

Testing of samples under study for potential use as the basis for lithium-ion batteries was carried out using two CR20 32 electrode cells and CT-3008W-5V equipment (Neware Technology Limited, Zhongkang Rd., Shenzhen, China). The test method and the description of the electrolyte are presented in [43].

The study of the dynamics of resizing as a result of annealing was carried out using the optical diffraction method M3-PALS technology Zetasizer Nano ZS (Malvern Instruments, Malvern, UK).

## 3. Results

### 3.1. Changes in the Morphology and Elemental Composition of Nanoparticles as a Result of Thermal Annealing

As is known, the method of chemical synthesis is the simplest and most scalable method of obtaining various nanoscale particles by mixing the initial components of the solution, followed by initiating processes of particle coagulation. Also, in contrast to mechanical chemical synthesis or sol-gel methods, during chemical synthesis, the sizes of the resulting initial nanoparticles can be varied by changing the concentration of solutions or other external factors. However, the nanoparticles obtained by this method in most cases are X-ray amorphous, due to the small size or the presence in the structure of a considerable number of vacancy defects or areas of disorder. Moreover, reducing the concentration of defects in nanoparticles, as well as changing the phase composition, can be achieved by a relatively simple method of thermal annealing of the nanoparticles obtained in an inert or oxygen-containing medium. The combination of these two methods of synthesis allows to obtain nanoparticles with a high crystallinity degree and a given phase composition and structural and magnetic properties. Figure 1 represents the dynamics of nanoparticle changes during thermal annealing, obtained using the method of scanning electron microscopy (SEM).

As is evident from the data provided, the initial particles are spherical particles whose average size does not transcend 20–25 nm. In this case, the presence of a magnetic phase in the structure results in the formation of agglomerates of particles. During heat treatment of nanoparticles at a temperature of 400 °C, there is a slight increase in particle size up to 25–27 nm. As the temperature of annealing increases to 600 °C, the particles grow to 30–37 nm, and further thermal annealing result in the formation of large feather-like agglomerates, which presence is caused by sintering processes and subsequent phase transformations under temperature. The determination of the size of nanoparticles was carried out by analyzing the images of raster and transmission electron microscopy, as well as their comparison. To determine the average size of nanoparticles, the estimation was carried out by determining the particle size using the ImageJ program code followed by the construction of Gaussian distributions. Figure 2a presents the results of the average size and measurement error obtained as a result of data analysis.

Figure 2b presents the results of thermo gravimetric analysis (TGA) of nanoparticles under study during thermal annealing. As could be observed from the data represented in annealing, a decrease in the mass of nanoparticles is observed. That is caused by a change in the phase composition and burn up of amorphous inclusions containing oxygen. The results of the change in the atomic ratio of the components as a result of the annealing are presented in Table 1. The data were obtained using the method of energy dispersive analysis by taking spectra from different parts of nanoparticles and determining the average values of parameters. Also, studies of the uniformity of the distribution of elements were estimated by taking maps of the distribution of elements using the mapping method.

A rise in the annealing temperature result in a slight reduction in the oxygen content in nanoparticles structure and an increase in the iron content, which indicates the formation of new phases as an outcome of heat treatment. Figure 3 shows the dependences of the change in the ζ-potentials of nanoparticles before and after thermal annealing in media with diverse pH.

The isoelectric point for initial nanoparticles is observed at pH = 6.5. In this case, the surface charge density is insufficient to prevent nanoparticles from agglomerating and forming large nanoparticle stability agglomerations. As an outcome of thermal annealing and the formation of CeFeO_3_ and Fe_2_O_3_ phases in the nanoparticle structure, the isoelectric point shifts towards pH = 4. That indicates an increase in inertness to self-organization and conglomeration of nanoparticles. The increase in inertia is connected to the formation of new phases in the structure and the emergence of a magnetic texture characteristic of hematite.

### 3.2. Studies of Phase Transformations in Nanoparticles as a Result of Thermal Annealing

One of the important characteristics of studied nanoparticles is the change in crystallographic characteristics and phase composition as a result of thermal heating. The most credible approach for evaluating these changes is the X-ray phase analysis method. Figure 4 represents the dynamics of changes in X-ray diffraction patterns in the annealing process. According to the diffractograms obtained, the samples under study are nanoscale polycrystalline objects, as evidenced by the broadened shape of the diffraction maxima and low intensity.

According to the X-ray diffraction patterns obtained, the phase composition of the initial nanoparticles was determined. As is known, in the case of a chemical method for producing nanoparticles based on iron salts, the preferred form of nanoparticle formation is the X-ray amorphous phase of magnetite iron oxide Fe_3_O_4_, which is formed as an outcome of chemical reactions. However, the addition of cerium nitrate to the initial solution leads to the creation of a crystalline phase in the structure. It was established that the structure of initial nanoparticles is characteristic of the cubic phase of cerium oxide CeO_2_, which confirms measurements of the atomic ratio of components in the structure. The low intensity, as well as the broadened and asymmetrical shape of diffraction maxima for the original nanoparticles, is a consequence of the presence of amorphous-like inclusions of iron oxide in the structure. That corresponds to low-intensity and highly asymmetric peaks in the region of 50–70° and a high oxygen share in the structure. A growth in the temperature of annealing gives rise to an increase in intensities and a narrowing of the diffraction maxima. That indicates a drop in contributions of amorphous inclusions and an increase in the degree of structure ordering. Table 2 represents the outcome of the evaluation of the phase composition during heat treatment. Evaluation of alterations in the phase composition in the structure of nanoparticles was conducted using the Rietveld method. The determination of volume fraction of the phase contribution was conducted with the use of Formula (1):(1)$$V_{admixture}=\frac{RI_{phase}}{I_{admixture}+RI_{phase}}$$

*R* is the structural coefficient equal to 1.45, *I_phase_* is the average integrated intensity of main phase of the diffraction line, *I_admixture_* is the average integrated intensity of the additional phase [44]. Table 2 presents data on changes in various contributions depending on the heat treatment temperature.

According to the data presented in Table 2, it is evident that the main phase transformations occur at a temperature of 600 °C when the formation of the orthorhombic phase CeFeO_3_ and the rhombic phase of hematite is observed in the structure of nanoparticles. In this case, there is a narrowing of diffraction peaks and an increase in intensity, which indicates amelioration in the structure of the crystals and a drop in amorphous inclusions. The emergence in the structure of nanoparticles of peaks characteristic of the rhombic phase of the hematite Fe_2_O_3_ at an annealing temperature of 600 °C is due to the fact that according to the phase transition diagram, the transition from the magnetite phase to hematite occurs in the vicinity of 500–600 °C. Therefore, it can be assumed that the formed X-ray amorphous phase of magnetite during the synthesis process, during thermal annealing, is transformed into the hematite phase with its subsequent crystallization. Moreover, the presence of the orthorhombic phase CeFeO_3_ may be due to the onset of crystallization of the hematite phase with a high content of cerium in the structure of the crystal lattice, or crystallization and replacement of cerium atoms in the lattice sites by iron heating as a result of thermal heating. The formation of the magnetic phase of hematite in nanoparticles structure at a temperature of 600 °C is consistent with previous experiments and the literature data on the phase transformations of iron-based oxide nanoparticles [41,42,43,44]. Also, at a temperature of 800 °C, the structure of nanoparticles exhibits the presence of two phases of cerium oxide and hematite in a ratio close to 2:1.

A growth in the annealing temperature results in a rise in the hematite phase. Table 3 and Table 4 represent the alterations in the principle crystallographic characteristics.

In accordance with the presented data, a rise in the lattice parameters is attributable to an alteration in the concentration of elements in the structure, the processes of introduction and substitution of atoms in the lattice sites under thermal annealing, as well as the formation of new phases and a drop in regions of disorder (results of Table 4). A rise in the temperature of annealing contributes to the migration and subsequent removal of oxygen from the structure of nanoparticles. That results in ordering of the crystal lattice with the following formation of the magnetic phase of hematite in the structure. At an annealing temperature of 800 °C, the content of amorphous inclusions in the structure does not transcend 13%, which demonstrates a high degree of orderliness of nanoparticles. A decrease in dislocations density, which gives rise to a decrease in dislocation defects and distortions in the structure, also indicates a drop in amorphous inclusions and an ordering of the crystal structure. Figure 5 represents transmission electron microscope (TEM) images of initial nanoparticles and after heat treatment at 800 °C to confirm the presence of amorphous inclusions and regions of disorder in the structure.

In compliance with the TEM images, the presence of amorphous regions in the structure of the initial particles indicates the formation of X-ray amorphous phase characteristic of iron oxide. In the case of heat treatment, there is a drop in amorphous and disordered inclusions in the structure of particles, as well as an alteration in their shape from hexagonal to spherical.

### 3.3. The Study of Changes in the Magnetic Texture of Nanoparticles as a Result of Thermal Annealing

The effect of thermal annealing on the change in magnetic characteristics of investigated nanoparticles was conducted applying Mossbauer spectroscopy approach. The dynamics of changes in Mossbauer spectra is shown in Figure 6. The determination of texture parameters was conducted based on the analysis of Mössbauer spectra [37,43]. That enables us to estimate parameters correlations, as well as the range of their alterations.

For initial nanoparticles, the Mössbauer spectrum is a line of noise measurements with low-intensity peaks characteristic of an unformed magnetic phase. For specimens annealed at a temperature of 400 °C, an asymmetric doublet is observed, which is characteristic of the paramagnetic state of iron oxide FeO in the amorphous phase. Also, the strong broadening and asymmetry of lines indicates a strong disordering and the presence in the structure of a high content of cationic vacancies, which reaffirms the X-ray phase analysis data. Increasing the annealing temperature to 600 °C, a narrowing of the doublet lines and the emergence of sextet lines characteristic of the Fe_2_O_3_ phase is observed. The magnitude of the hyperfine magnetic field for a sextet is 503.7 kOe, which is consistent with a field characteristic of the hematite phase in a highly disordered state. At an annealing temperature of 800 °C, the Mössbauer spectrum is a sextet characteristic of hematite (the magnitude of the hyperfine magnetic field is 512.5 kOe) and a low-intensity paramagnetic doublet characteristic of amorphous inclusions [37,41]. The deviation of the magnitude of the hyperfine magnetic field for the nanoparticles under study from the reference value is due to the presence of impurity phase inclusions in the structure, as well as regions of structure disorder caused by the presence of different phases. According to the data obtained, a growth in the annealing temperature result in a drop in the contribution of the partial spectrum characteristic of the paramagnetic state, which demonstrates a drop in the concentration of cationic vacancies in the crystal structure and its ordering. Also, the formation of a sextet characteristic of the hematite phase indicates the appearance of a magnetic texture in the structure of nanoparticles with its further ordering [37,41].

### 3.4. The Study of the Optical Properties of Nanoparticles

One of the important characteristics of obtained nanoparticles are optical characteristics and their change as a result of phase transformations [45]. Figure 7 shows the change in the optical absorption spectra of the researched nanoparticles before and after annealing.

The UV spectra of the nanoparticles under study have a broad absorption from 300 to 1100 nm. Changes in the UV spectra from the initial ones are observed for samples calcined at 800 °C and 1000 °C. There is a change in absorption at 378 nm and a slight shift in the peak from 547 to 574 nm.

### 3.5. Investigation of the Corrosion Resistance

Essential characteristics of the corrosion resistance of nanostructures are the oxidation reaction rate and following corrosion, which entails an alteration in the concentration of substances per unit of time [46]. The results on the corrosion resistance of nanoparticles in acid solutions are presented in order to determine the oxidation products and the rate of degradation. The choice of acids for the experiment to determine corrosion resistance was based on the fact that most electrolytes use slightly soluble acids, and if their concentration increases dramatically, nanoparticles can quickly degrade due to the accelerated oxidation process. A highly concentrated solution of sulfuric acid, which is used in most electrolytes, was used as an aggressive medium. Figure 8 shows the kinetic curves of researched specimens.

As is evident from the represented data, the alteration in anamorphosis in coordinates $\ln\frac{\left[A_{0}\right]}{\left[A_{0}-x\right]}-t$, where *A*_0_ is the initial concentration of a substance, *A*_0_ − *x* is the current concentration of a substance at a given time, *t* is time, is described by a straight line, which indicates that the degradation processes of nanoparticles are first-order reactions [46]. In this case, a decrease in amorphous inclusions and a decline in the oxygen content in the structure as a consequence of thermal annealing result in a drop in degradation rate of nanoparticles. Also, the formation of a new phase of hematite, which prevents the corrosion of nanoparticles as a consequence of interaction with aggressive media, has an impact on reducing the degradation rate.

### 3.6. Resource Testing of Ceramics as Cathode Materials for Lithium-Ion Batteries

The principal distinctive features of lithium-ion batteries are high energy density, fast charging process, long service life (above 500 discharge/charge cycles) [43,44]. Tablets pressed from nanoparticles in a weight ratio of 1 g were used as test samples. Electrochemical testing of synthesized and annealed ceramics was carried out in two-electrode CR20 32 cells on a charge-discharge test bench CT-3008W-5V (Neware company, HC Tower, No. 1 Sheung Yuet Rd, Kowloon Bay, Hong Kong). Metallic lithium was used as a counter electrode. Electrochemical testing of copper nanostructures was carried out in two-electrode CR20 32 cells on a charge-discharge test bench CT-3008W-5V (Neware company, HC Tower, No. 1 Sheung Yuet Rd, Kowloon Bay, Hong Kong). Metallic lithium was used as a counter electrode. The electrolyte solution was a mixture of 1 M LiPF6, 1 M ethylene carbonate, 1 M propylene carbonate, 0.1 M diethyl carbonate, 1 M ethyl methyl carbonate, 1 M propyl acetate (TC-E810 Tinci, Neware company, HC Tower, No. 1 Sheung Yuet Rd, Kowloon Bay, Hong Kong) [43]. Anode cycling was carried out in galvanostatic mode in the voltage range from 10 mV to 2 V, in the mode of limiting the charging capacity of 1000 mA h/g. In the anode cycle, limiting was the achievement of 2 V voltage. Figure 9a represents the outcome of endurance tests of the researched specimens.

As is evident from the represented data, the decrease in amorphous inclusions in the structure result in a rise in the number of cycles of endurance tests at which the specific discharge capacity does not decrease below 80%. Figure 9b presents a graph of the lifetime with the storage capacity exceeding 80% for the initial and modified nanostructures when cycling in the charge capacity mode of 1000 mA h/g. An increase in the lifetime for samples annealed at 600 °C and 800 °C is due to the formation of a hematite phase in the structure, which leads to a rise in crystallinity and a drop in amorphous inclusions in the structure, which have a negative effect on the degradation rate of the electrode material. Figure 9c shows the SEM images of the studied nanoparticles after life tests. The decrease in battery life associated with a decrease in capacity below 80% of the initial value is due to the degradation of structural properties due to an increase in the content of amorphous-like inclusions and partial degradation of nanoparticles as a result of lithiation processes. Since the main processes of lithiation occur through reactions involving the exchange of oxygen, which leads to a change not only in the oxygen concentration, but also in partial degradation of the structure as a result of breaking chemical and crystalline bonds in the lattice. Figure 10 shows the dynamics of changes in the degree of crystallinity of the studied nanoparticles before and after the tests.

As can be seen from the data presented, after the endurance tests, a sharp decrease in the degree of crystallinity is observed, which indicates an increase in the structure of amorphous-like inclusions and regions of disorder. At the same time, the greatest decrease in the degree of crystallinity is stipulated for samples whose structure initially contained a high concentration of dislocation defects and disordered regions, while for samples annealed there is less change in the degree of crystallinity, which leads to an increase in the lifetime of nanostructures.

Figure 11 shows the charging curves of the first three cycles of the researched specimens before and after annealing [43,44]. The cycling was conducted in the galvanic mode in the voltage range (U) from 10 mV to 3 V.

As is evident from the represented data for specimens with the formed phase of iron oxide in the structure, a rise in the charging capacity is observed. This increase is attributable to the fact that the principal lithiation processes take place on iron oxide according to the following conversion reaction:6Li + Fe_2_O_3_ => 2Fe + 3Li_2_O

The presence in the structure of electrode material of a significant number of amorphous inclusions leads to a sharp degradation of the material and a decrease in the number of operating cycles. It should be noted that the degree of oxidation of Fe completely changes when the electrode is lithiated and divisible, which result in subsequent degradation of the electrode material. The difference between the charging curve of the first cycle and the following ones is attributable to the slight amorphization of the electrode material.

## 4. Conclusions

The article represents the outcome of the production of ferrite nanoparticles and the further evolution of structural and optical properties as a consequence of thermal annealing. The choice of synthesis conditions and temperature range is due to phase transformations according to state diagrams, as well as the possibility of obtaining CeO_2_-Fe_2_O_3_ nanoparticles. As a result of the research, it was found that chemically synthesized nanoparticles are highly disordered polycrystalline structures of cerium oxide, with the presence of an amorphous phase of iron oxide in the structure. During thermal annealing of nanoparticles in an oxygen-containing medium, the formation of new phases containing iron oxides and the following ordering of the crystal structure and magnetic texture is observed. At an annealing temperature over 600 °C, the formation of an ordered magnetic phase of hematite is observed, which result in the appearance of magnetic properties of nanoparticles. The appearance of the hematite phase is a consequence of phase transformations of the oxide forms of iron from amorphous magnetite to hematite with the formation of the transition phase CeFeO_3_. In this case, with a growth in the annealing temperature, a decline in the regions of disorder and a decrease in the number of dislocation defects are observed, which is due to the crystallization processes and the emergence of new phases. In accordance with the data of Mössbauer spectroscopy, it was established that a rise in the annealing temperature result in an ordering of the magnetic texture and a decrease in the concentration of cationic and vacancy defects in the structure. It was established that a decrease in amorphous inclusions and a decrease in the oxygen content in the structure as a result of thermal annealing result in a decrease in the degradation rate of nanoparticles. The prospects of using the nanoparticles obtained as the basis of electrode materials for lithium-ion batteries are shown. It was revealed that the emergence of hematite in the phase structure results in a rise in the working life of lithium-ion batteries.

## Figures and Tables

**Figure 1 nanomaterials-09-01079-f001:**
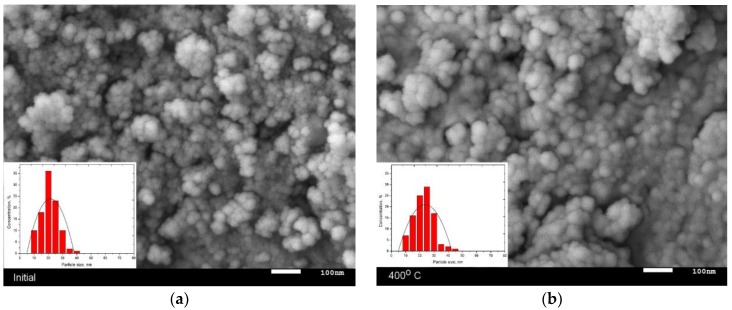
Dynamics of change in the geometric parameters of nanoparticles according to scanning electron microscopy (SEM) image data: (**a**) initial; (**b**) 400 °C; (**c**) 600 °C; (**d**) 800 °C.

**Figure 2 nanomaterials-09-01079-f002:**
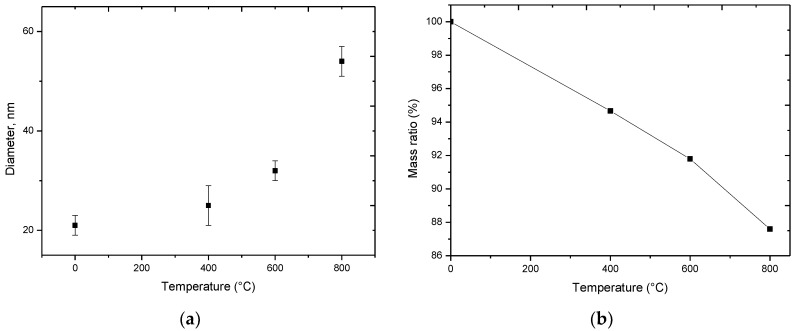
(**a**) Dynamics of change in the average size of nanoparticles; (**b**) results of thermo gravimetric analysis (TGA) change in mass of studied nanoparticles as a result of annealing.

**Figure 3 nanomaterials-09-01079-f003:**
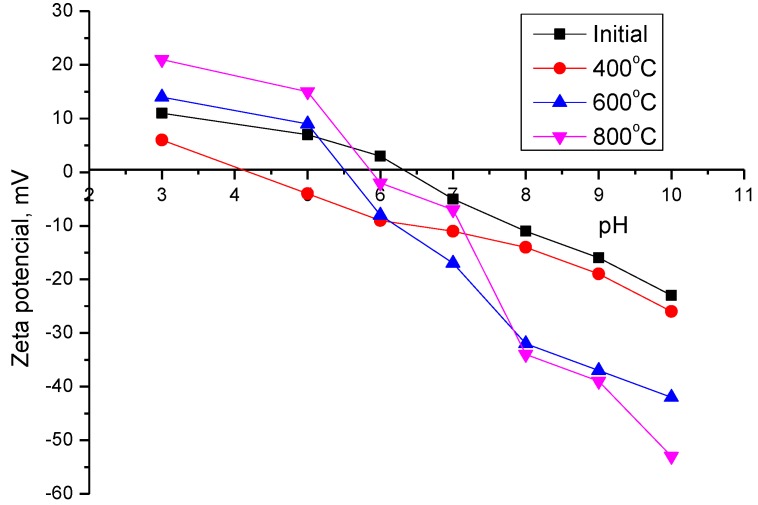
Dynamics of changes of zeta potentials in the process of nanoparticles annealing.

**Figure 4 nanomaterials-09-01079-f004:**
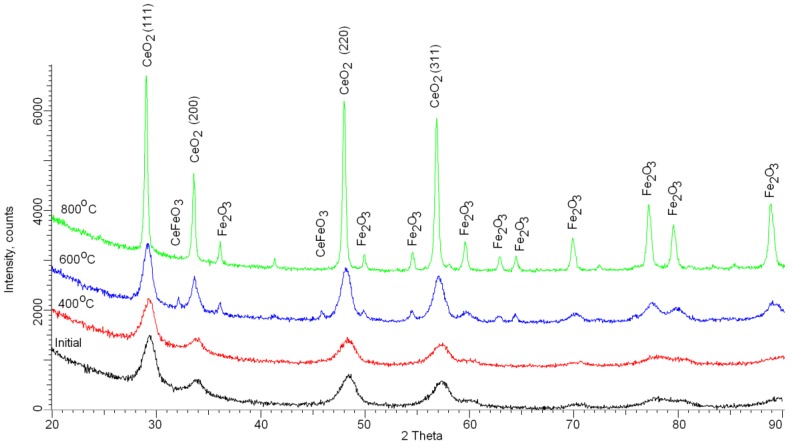
X-ray diffraction (XRD) patterns of the studied nanoparticles before and after annealing.

**Figure 5 nanomaterials-09-01079-f005:**
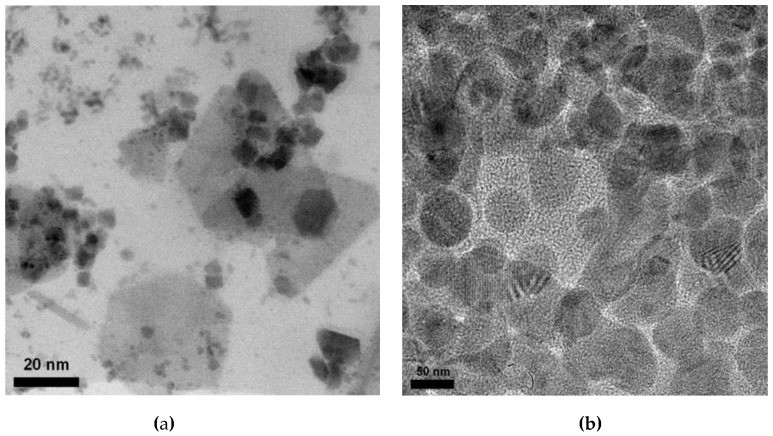
Transmission electron microscope (TEM) images of initial nanoparticles and after heat treatment: (**a**) Initial; (**b**) 800 °C.

**Figure 6 nanomaterials-09-01079-f006:**
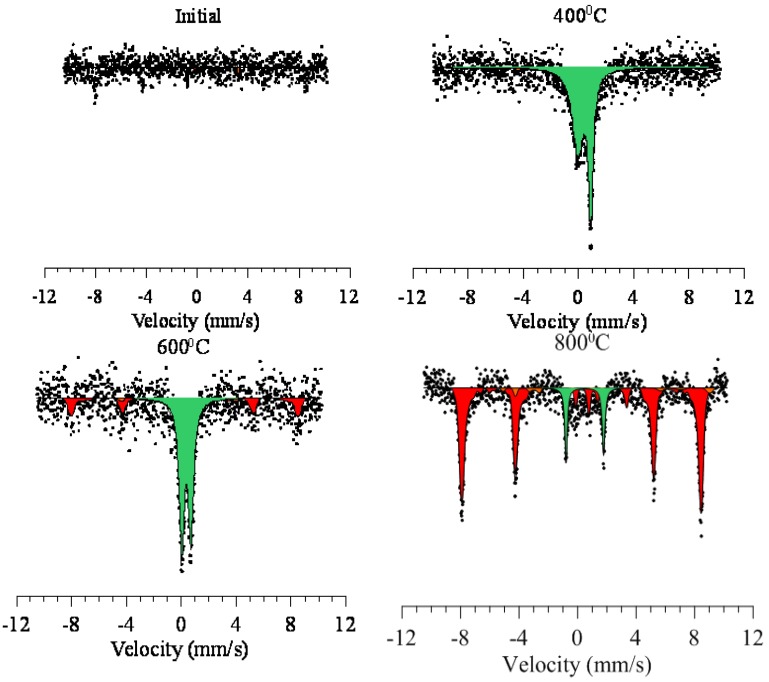
Mössbauer spectra of the studied nanoparticles before and after annealing.

**Figure 7 nanomaterials-09-01079-f007:**
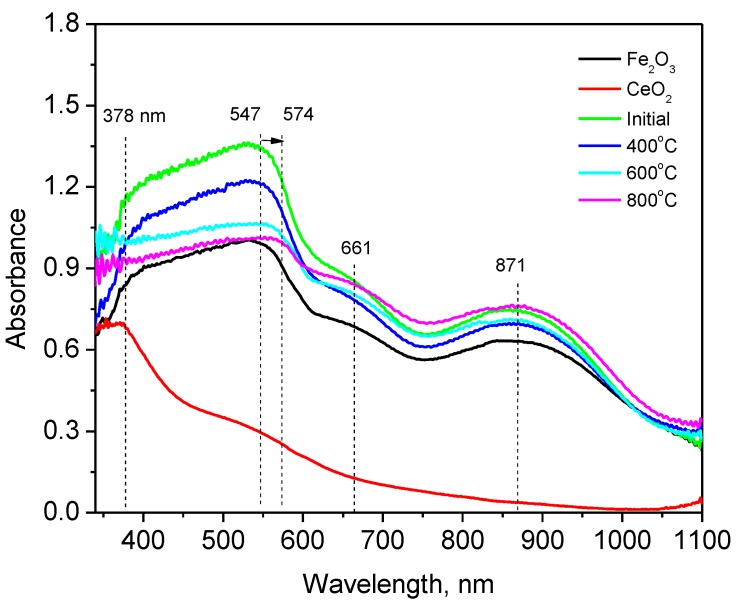
Dynamics of change in the absorption spectrum of studied nanoparticles.

**Figure 8 nanomaterials-09-01079-f008:**
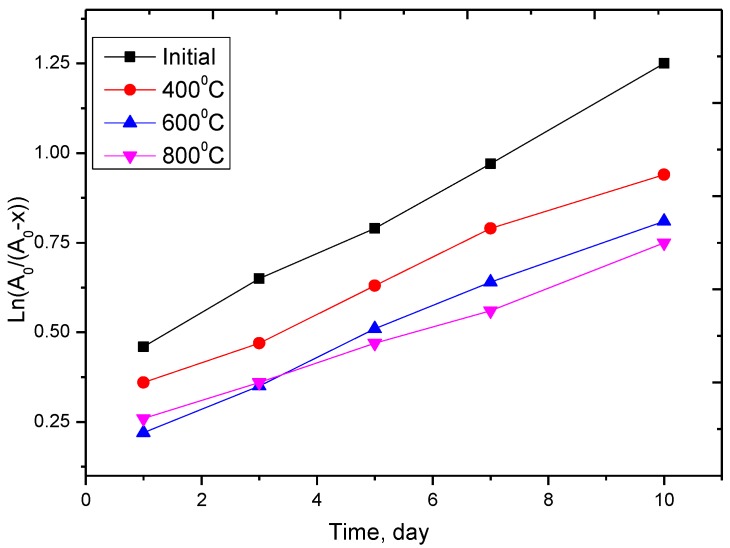
Anamorphosis of the kinetic curve for the oxidation reaction and degradation of investigated specimens.

**Figure 9 nanomaterials-09-01079-f009:**
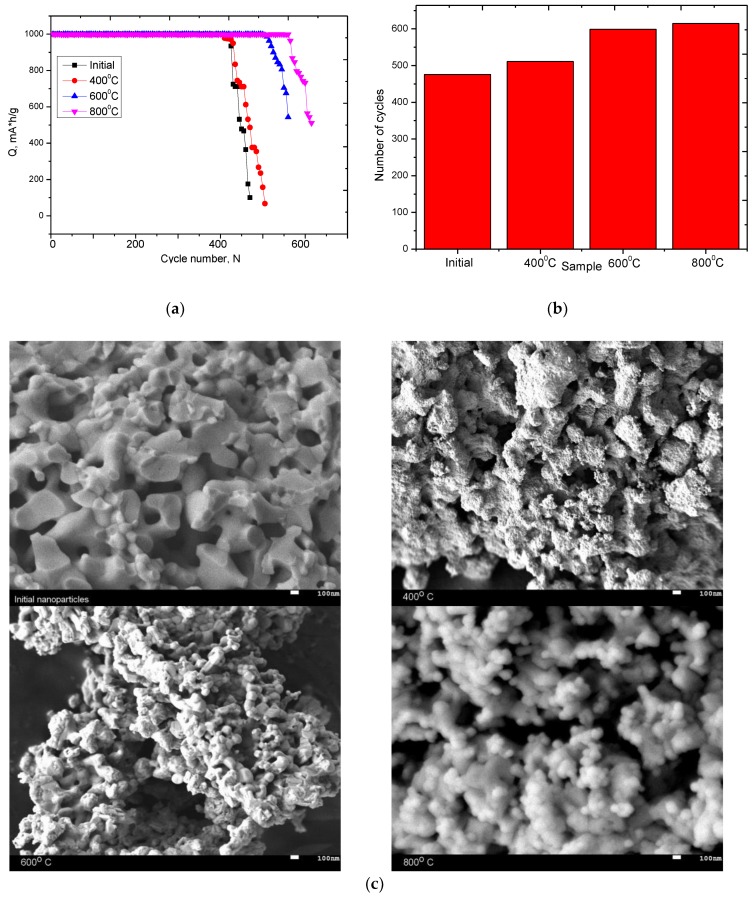
(**a**) Graph of specific discharge capacity versus the number of cycles tested in charge capacity mode 1000 mA h/g; (**b**) graph of the lifetime of nanoparticles on the type of modification (before degradation begins and capacity decreases below 80%); (**c**) SEM images of the studied nanoparticles after life tests.

**Figure 10 nanomaterials-09-01079-f010:**
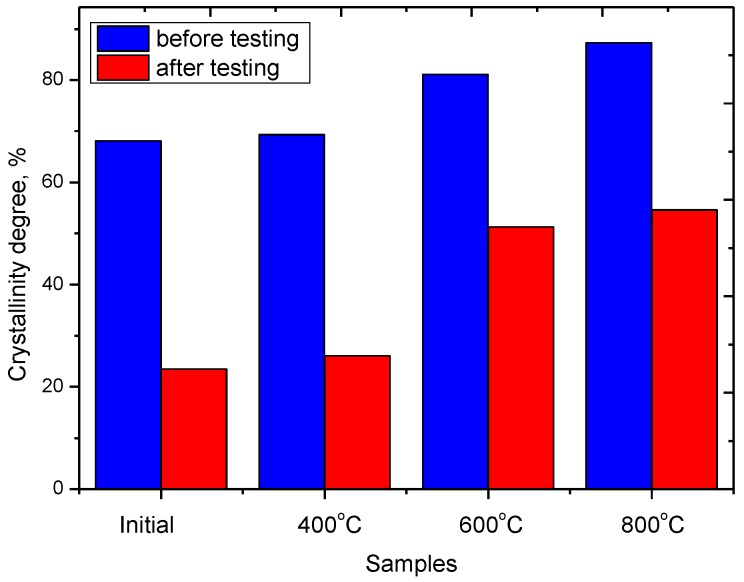
The dynamics of changes in the degree of crystallinity before and after life tests.

**Figure 11 nanomaterials-09-01079-f011:**
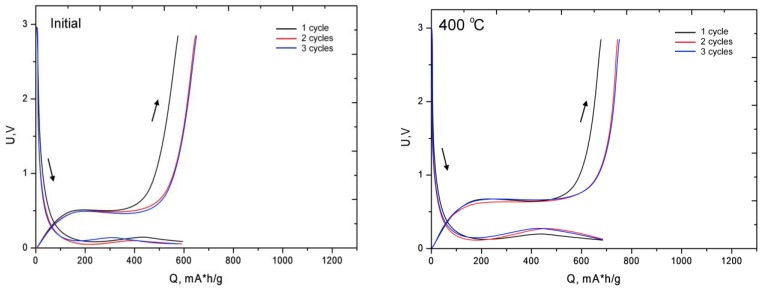
Charge-discharge curves of the first 3 cycles. The descending arrows demonstrate the charge (introduction of lithium into the anode), ascending demonstrate discharge (lithium extraction).

**Table 1 nanomaterials-09-01079-t001:** Elemental composition data of studied nanoparticles.

Sample	Atomic Ratio, %		
	Ce	Fe	O
Initial	59.3 ± 1.3	16.1 ± 1.1	24.6 ± 1.5
400 °C	54.6 ± 1.5	20.3 ± 1.2	21.1 ± 1.4
600 °C	59.8 ± 1.6	20.4 ± 1.4	19.8 ± 1.7
800 °C	59.6 ± 1.5	22.6 ± 1.2	17.8 ± 1.3

**Table 2 nanomaterials-09-01079-t002:** Data of phase composition.

Phase	Type of Structure	Space Group	Phase Content, %			
			Initial	400 °C	600 °C	800 °C
CeO_2_—Cerianite	Cubic	Fm-3m (225)	100	100	66.6	65.1
CeFeO_3_	Orthorhombic	Pbnm (62)	-	-	23.7	-
Fe_2_O_3_—Hematite	Rhombo.H.axes	R-3c (167)	-	-	9.7	34.9

**Table 3 nanomaterials-09-01079-t003:** Data on alterations in parameters of the crystal lattice.

Phase	Lattice Parameter, Å			
	Initial	400 °C	600 °C	800 °C
CeO_2_—Cerianite	a = 5.2761	a = 5.2803	a = 5.2917	a = 5.3197
CeFeO_3_	-	-	a = 5.6151, b = 5.7473, c = 7.8581	-
Fe_2_O_3_—Hematite	-	-	a = 4.9756, c = 13.7846	a = 4.9432, c = 13.7552

**Table 4 nanomaterials-09-01079-t004:** Data on alterations in the main crystallographic characteristics.

Sample	Crystalline Size, nm	Crystallinity Degree, %	Dislocation Density, 10^15^
Initial	6.7 ± 0.8	68.1 ± 2.2	22.27
400 °C	7.5 ± 1.2	69.3 ± 3.1	17.78
600 °C	10.9 ± 1.4	81.1 ± 3.6	8.42
800 °C	28.2 ± 2.1	87.3 ± 3.4	1.26

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
