# Peer review of "Synthesis and Properties of Ferrite-Based Nanoparticles"

_nanomaterials, 2019, doi:10.3390/nano9081079_

Reviewer 1 Report

To my opinion, even though this manucript presents a set of interesting data, it should not be accepted in the present form. Below, please find my comments substantiating this assessment:

No information on the stoichiometry, nor the concentration/pH control of the synthetic route is provided. This is important not only because it would be difficult to follow the procedure for those interested, but because inappropriate adjustments in pH  can result in mixed oxide/hydroxide formation.

The formation of mixed hydroxides as a final precipitation product can explain further temperature annealing, by the formation of oxides.

Generally, hematite is considered non-magnetic. If the Authors claim the appearance of magnetic properties/textures of hematite after annealing, this should be supported by direct magnetic measurements with either VSM or SQUID techniques (cf.: a series of papers by Tadic M., et al, Appl. Surf. Sci. 320, 183, 2014 and more recent, up to 2019).

Zeta potentials should be reported and compared ONLY under the same ionc strength  conditions. The position of the plane of shear and the value of zeta potential strongly depend upon the ionic strength. pH titration affects the ionic strength, if not compensated by an inert salt.

Fig.7. Since the spectra are not normalized and a huge scatter is observed in the UV region, it is very difficult to follow the reasoning of the Authors. Besides, the absorption region between 800 and 1000 nm is quite different thatn that of Fe2O3, contrasting the claim of the last sentence of the Authors in chapter 3.4.

Minor comment: too many references (56) as for the research article of this length

Overall, the manuscript requires thorough improvements before it can be reconsidered.

Author Response

1. Ce(NO3)3*6H2O, FeCl3*6H2O, HCl and NaOH were used as the source components for the chemical synthesis of ferrite nanoparticles based on iron and cerium oxides. Ce (NO3)3*6H2O (17.36 g) was dissolved in a solution of 400 ml of dioinized water + 1 ml of HCl, then a mixture of FeCl3*6H2O (10.8  g) + 400 ml of dioinized water + HCl 1 ml was added, the solution was aged for 48 hours in argon medium, after which the precipitate was separated by centrifuging, washed and dried in argon medium. The pH control was carried out using a pH meter and brought to a level of 9.3 by adding drops of an alkaline solution of NaOH. At this pH level, the formation of hydroxide inclusions is excluded, which was also confirmed using x-ray phase analysis, according to which hydroxide structures are not observed in the structure of the initial samples. Annealing of synthesized nanoparticles was conducted in an oxygen-containing medium in a muffle furnace at a temperature of 200, 400, 600 and 800 °C for 5 hours.

2. The presence of the magnetic structure of hematite and its texture (i.e., the presence of a dedicated axis of magnetic domains) was determined using the Mössbauer spectroscopy method, which makes it possible, with a high degree of probability, to separate the contributions from the various magnetic components of iron oxides, and also to determine the phase state of iron oxide by estimating magnitudes of ultrafine magnetic fields. In this work, the magnitude of the hyperfine magnetic field for the observed sextet was 512.5 kOe, which is close enough to the value for the ordered hematite phase (Hn = 517 kOe). The deviation of the magnitude of the hyperfine magnetic field for the nanoparticles under study from the reference value is due to the presence of impurity phase inclusions in the structure, as well as regions of structure disorder caused by the presence of different phases.

I would also like to note that at the moment our team is planning to study the magnetic characteristics of the particles under study, and also to change them under the influence of external factors using the VSM or SQUID techniques at the end of 2019. The results of these studies will be published in our subsequent work. At the moment, our team does not have the opportunity to conduct such research.

3. The results of the study of the zeta potential for the obtained nanoparticles were carried out under the same conditions. The dependences obtained were constructed to determine the isoelectric point, the value of which shows the ability to agglomerate nanoparticles.

4. The text of the article corrected.

Figure 7. Dynamics of change in the absorption spectrum of studied nanoparticles.

The UV spectra of the nanoparticles under study have a broad absorption from 300 to 1100 nm. Changes in the UV spectra from the initial ones are observed for samples calcined at 800 ° C and 1000 ° C. There is a change in absorption at 378 nm and a slight shift in the peak from 547 to 574 nm

Reviewer 2 Report

Dear Editor,

The authors presented the results concerning the properties of system CeO2 - Fe2O3, not the CeFeOx. The CeFeO3 presented only in one sample with the other phases of CeO2 and Fe2O3. The title is not correct. The manuscript cannot be published in the present form and should be improve.

My remarks are:

1.     In the Abstract was mention “The work is dedicated to the study of structural, optical and magnetic characteristics, as well as phase transformations of ferrite nanoparticles of CeFeOx.” The authors did not show the magnetic characteristics.

2.     What was the method of synthesis? Please, give the detailed information for the synthesis process.

3.     Give briefly description of the measurement procedure of Moesbauer spectroscopy measurements and optical propertiesр nevertheless they were published.

4.     How were the elemental composition data obtained shown in Table 1?

5.     How was the oxygen contains determine in particles?

6.     How was the particle size determine and what was the particle size distribution?

7.     The magnetite are very sensible to the oxygen. You did not mention that you synthesized the initial samples at inert atmosphere. Are you sure that the amorphous phase is not the Fe2O3 or iron (hydro)oxidies?

8.     The authors wrote “the transition 184 from the magnetite phase to hematite occurs in the vicinity of 500-600°C.” (Line 184). It is not correct. It is well known that the transition from the magnetite phase to hematite can be occur at temperature above 100°C as the Fe2+ is easy oxidized to Fe3+ in the air even at room temperature.

9.     The temperature of 600°C is not enough to obtain the single phase of CeFeOx, but why is CeFeO3 disappear at 800°C? Do you have any explanation?

10.  How the data presented in Table 3 and Table 4 were obtain?

11.  There is not enough information how the CeFeO3 influence on corrosion resistance and the test of obtained samples as cathode materials.

12.  The authors don’t described in what type form – powder or bulk, the samples for specific discharge capacity versus the number of cycles measurements were used.

13.  On Figure 9b the authors presented “Graph of the lifetime of ceramic on the type of modification”, but there is not information for the microstructure and homogeneity of the phase’s dispersion in the bulk samples. The density of the bulk also should be take into consideration.

14.  In the conclusion was wrote (Line 311-315) “During thermal annealing of nanoparticles in an oxygen-containing medium, the formation of new phases containing iron oxides and the following ordering of the crystal structure  and magnetic texture is observed. At an annealing temperature over 600°C, the formation of an ordered magnetic phase of hematite is observed, which result in the appearance of magnetic properties of nanoparticles.”. It was not shown the magnetic texture. What do you mean?

I suggest the manuscript to be rewrite and send again to the journal.

Author Response

1. In this work, the magnetic properties, the presence of the magnetic phase of hematite, in particular, and the determination of the hyperfine parameters of the magnetic characteristics were carried out using the Mössbauer spectroscopy method.

The abstract has been corrected, the word magnetic characteristics has been removed.

2. Ce(NO3)3*6H2O, FeCl3*6H2O, HCl and NaOH were used as the source components for the chemical synthesis of ferrite nanoparticles based on iron and cerium oxides. Ce (NO3)3*6H2O (17.36  g) was dissolved in a solution of 400 ml of dioinized water + 1 ml of HCl, then a mixture of FeCl3*6H2O (10.8 g) + 400 ml of dioinized water + HCl 1 ml was added, the solution was aged for 48 hours in argon medium, after which the precipitate was separated by centrifuging, washed and dried in argon medium. The pH control was carried out using a pH meter and brought to a level of 9.3 by adding drops of an alkaline solution of NaOH. At this pH level, the formation of hydroxide inclusions is excluded, which was also confirmed using x-ray phase analysis, according to which hydroxide structures are not observed in the structure of the initial samples. Annealing of synthesized nanoparticles was conducted in an oxygen-containing medium in a muffle furnace at a temperature of 200, 400, 600 and 800 °C for 5 hours.

3. The Mossbauer spectrometer was calibrated at room temperature using a standard α-Fe absorber. For processing and analyzing the Mossbauer spectra, methods were used to restore the distributions of the hyperfine parameters of the Mossbauer spectrum, taking into account a priori information about the object of study, implemented in the SpectrRelax program.

Diffuse reflectance UV-Vis spectra were recorded using Analytic Jena Specord-250 BU spectrophotometer equipped with integrating sphere. BaSO4 was used as a standard. The resolution was chosen to be 1 nm and the scan speed was 20 nm/s. The spectral range was from 190 nm to 1100 nm.

4-5. The data were obtained using the method of energy dispersive analysis by taking spectra from different parts of nanoparticles and determining the average values of parameters. Also, studies of the uniformity of the distribution of elements were estimated by taking maps of the distribution of elements using the mapping method.

6. The determination of the size of nanoparticles was carried out by analyzing the images of raster and transmission electron microscopy, as well as their comparison. To determine the average size of nanoparticles, the estimation was carried out by determining the particle size using the ImageJ program code followed by the construction of Gaussian distributions. Figure 2a presents the results of the average size and measurement error obtained as a result of data analysis.

7. The synthesis was carried out in an inert gaseous medium of argon. The presence of the magnetic structure of hematite and its texture (i.e., the presence of a dedicated axis of magnetic domains) was determined using the Mössbauer spectroscopy method, which makes it possible, with a high degree of probability, to separate the contributions from the various magnetic components of iron oxides, and also to determine the phase state of iron oxide by estimating magnitudes of ultrafine magnetic fields. In this work, the magnitude of the hyperfine magnetic field for the observed sextet was 512.5 kOe, which is close enough to the value for the ordered hematite phase (Hn = 517 kOe). The deviation of the magnitude of the hyperfine magnetic field for the nanoparticles under study from the reference value is due to the presence of impurity phase inclusions in the structure, as well as regions of structure disorder caused by the presence of different phases.

8. The authors agree with this statement that iron oxide is oxidized at temperatures above 100°C, but obtaining the phase of pure hematite with full phase transformation as it is known from the literature data for nanoparticles occurs at temperatures above 500-600°C. In this case, according to the data of Mössbauer spectroscopy, the presence of spectral lines characteristic of hematite is observed only at an annealing temperature above 600°C.

9. An increase in the annealing temperature leads to a decrease in amorphous-like inclusions in the structure of nanoparticles, as well as the formation of the hematite phase in the structure, the appearance of which is associated with the formation of the transition phase of a solid solution of iron intercalation in the structure of СеО2 with the formation of the CeFeO3 phase, a further increase in the annealing temperature to 800 ° C result in displacement of cerium from the lattice sites by iron and the formation of the hematite phase Fe2O3.

10. The data of crystallographic characteristics and the dynamics of their changes were obtained as a result of the analysis of X-ray diffractograms using the DiffracEVA 4.2 and TOPAS 4.0 software based on the Rietveld method. Methods for determining the parameters are presented in our previously published papers Kozlovskiy, A., et al. "Study of the use of ionizing radiation to improve the efficiency of performance of nickel nanostructures as anodes of lithium-ion batteries." Materials Research Express 6.5 (2019): 055026, Kozlovskiy, А., et al. "Effect of irradiation with C2+ and O2+ ions on the structural and conductive characteristics of copper nanostructures." Materials Research Express 6.7 (2019): 075072. Kozlovskiy, A., et al. "Optical and structural properties of AlN ceramics irradiated with heavy ions." Optical Materials 91 (2019): 130-137. Kozlovskiy, A., et al. "Structure and corrosion properties of thin TiO2 films obtained by magnetron sputtering." Vacuum 164 (2019): 224-232.

11. The results on the corrosion resistance of nanoparticles in acid solutions are presented in order to determine the oxidation products and the rate of degradation. The choice of acids for the experiment to determine corrosion resistance was based on the fact that most electrolytes use slightly soluble acids, and if their concentration increases dramatically, nanoparticles can quickly degrade due to the accelerated oxidation process.

12. Tablets pressed from nanoparticles in a weight ratio of 1 g were used as test samples. Electrochemical testing of synthesized and annealed ceramics was carried out in two-electrode CR20 32 cells on a charge-discharge test bench CT-3008W-5V (Neware company). Metallic lithium was used as a counter electrode. Electrochemical testing of copper nanostructures was carried out in two-electrode CR20 32 cells on a charge-discharge test bench CT-3008W-5V (Neware company). Metallic lithium was used as a counter electrode. The electrolyte solution was a mixture of 1 М LiPF6, 1 M ethylene carbonate, 1 M propylene carbonate, 0.1 M diethyl carbonate, 1 M ethyl methyl carbonate, 1 M propyl acetate (TC-E810 Tinci) [43]. Anode cycling was carried out in galvanostatic mode in the voltage range from 10 mV to 2 V, in the mode of limiting the charging capacity of 1000 mA•h/g. In the anode cycle, limiting was the achievement of 2 V voltage.

13. Figure 9c shows the SEM images of the studied nanoparticles after life tests. The decrease in battery life associated with a decrease in capacity below 80% of the initial value is due to the degradation of structural properties due to an increase in the content of amorphous-like inclusions and partial degradation of nanoparticles as a result of lithiation processes. Since the main processes of lithiation occur through reactions involving the exchange of oxygen, which leads to a change not only in the oxygen concentration, but also in partial degradation of the structure as a result of breaking chemical and crystalline bonds in the lattice. Figure 10 shows the dynamics of changes in the degree of crystallinity of the studied nanoparticles before and after the test tests.

Figure 10. The dynamics of changes in the degree of crystallinity before and after life tests

As can be seen from the data presented, after the endurance tests, a sharp decrease in the degree of crystallinity is observed, which indicates an increase in the structure of amorphous-like inclusions and regions of disorder. At the same time, the greatest decrease in the degree of crystallinity is stipulated for samples whose structure initially contained a high concentration of dislocation defects and disordered regions, while for samples annealed there is less change in the degree of crystallinity, which leads to an increase in the lifetime of nanostructures.

14. The presence of the magnetic structure of hematite and its texture (i.e., the presence of a dedicated axis of magnetic domains) was determined using the Mössbauer spectroscopy method, which makes it possible, with a high degree of probability, to separate the contributions from the various magnetic components of iron oxides, and also to determine the phase state of iron oxide by estimating magnitudes of ultrafine magnetic fields. In this work, the magnitude of the hyperfine magnetic field for the observed sextet was 512.5 kOe, which is close enough to the value for the ordered hematite phase (Hn = 517 kOe). The deviation of the magnitude of the hyperfine magnetic field for the nanoparticles under study from the reference value is due to the presence of impurity phase inclusions in the structure, as well as regions of structure disorder caused by the presence of different phases.

Round  2

Reviewer 2 Report

Dear Editor,

The authors answered to all of my questions and remarks. They did the appropriate changes in the manuscript. In the Introduction part, line 74-76, should be change according to the abstract and conclusion. It should be remove “magnetic properties”. As they didn’t obtained the single phase CeFeO3 I suggest to be write that they investigated the properties of the different phases in CeO2-Fe2O3 system.

I suggest the paper to be published after the above changes is done.

Best regards

Author Response

This article represents the outcome of synthesis and studies of the structural and optical properties of nanoparticles CeO2 - Fe2O3, obtained by chemical synthesis and further thermal annealing in an oxygen-containing medium.
